# Whole-Genome Sequence Analysis of Antibiotic Resistance, Virulence, and Plasmid Dynamics in Multidrug-Resistant *E. coli* Isolates from Imported Shrimp

**DOI:** 10.3390/foods13111766

**Published:** 2024-06-05

**Authors:** Kidon Sung, Mohamed Nawaz, Miseon Park, Jungwhan Chon, Saeed A. Khan, Khulud Alotaibi, Javier Revollo, Jaime A. Miranda, Ashraf A. Khan

**Affiliations:** 1Division of Microbiology, National Center for Toxicological Research, U.S. Food and Drug Administration, Jefferson, AR 72079, USA; mnawaz023@gmail.com (M.N.); miseon.park@fda.hhs.gov (M.P.); saeed.khan@fda.hhs.gov (S.A.K.); ashraf.khan@fda.hhs.gov (A.A.K.); 2Department of Companion Animal Health, Inje University, Gimhae 50834, Republic of Korea; alvarmar@naver.com; 3Saudi Food and Drug Authority (SFDA), Riyadh 13513, Saudi Arabia; kholoudsafar@gmail.com; 4Division of Genetic and Molecular Toxicology, National Center for Toxicological Research, U.S. Food and Drug Administration, Jefferson, AR 72079, USA; javier.revollo@fda.hhs.gov (J.R.); jaime.miranda@fda.hhs.gov (J.A.M.)

**Keywords:** whole-genome sequence, *E. coli*, imported shrimp

## Abstract

We analyzed antimicrobial resistance and virulence traits in multidrug-resistant (MDR) *E. coli* isolates obtained from imported shrimp using whole-genome sequences (WGSs). Antibiotic resistance profiles were determined phenotypically. WGSs identified key characteristics, including their multilocus sequence type (MLST), serotype, virulence factors, antibiotic resistance genes, and mobile elements. Most of the isolates exhibited resistance to gentamicin, streptomycin, ampicillin, chloramphenicol, nalidixic acid, ciprofloxacin, tetracycline, and trimethoprim/sulfamethoxazole. Multilocus sequence type (MLST), serotype, average nucleotide identity (ANI), and pangenome analysis showed high genomic similarity among isolates, except for EC15 and ECV01. The EC119 plasmid contained a variety of efflux pump genes, including those encoding the acid resistance transcriptional activators (*gadE*, *gadW*, and *gadX*), resistance-nodulation-division-type efflux pumps (*mdtE* and *mdtF*), and a metabolite, H1 symporter (MHS) family major facilitator superfamily transporter (*MNZ41_23075*). Virulence genes displayed diversity, particularly EC15, whose plasmids carried genes for adherence (*faeA* and *faeC-I*), invasion (*ipaH* and *virB*), and capsule (*caf1A* and *caf1M*). This comprehensive analysis illuminates antimicrobial resistance, virulence, and plasmid dynamics in *E. coli* from imported shrimp and has profound implications for public health, emphasizing the need for continued surveillance and research into the evolution of these important bacterial pathogens.

## 1. Introduction

The average U.S. seafood consumption in 2021 was 20.5 pounds per person, according to the National Fisheries Institute. Shrimp was the most popular choice, with individuals consuming 5.90 pounds per person [1]. However, much shrimp is imported from outside the U.S. In 2020, the U.S. imported approximately 6 billion pounds of seafood products, with shrimp comprising 27% of the total value of edible imports [2]. Although the Food and Drug Administration (FDA) has not approved the use of any antibiotics in U.S. shrimp aquaculture, other nations use various antibiotics in their shrimp aquaculture practices [3]. Multiple studies have demonstrated the presence of antimicrobial residues, including β-lactam, erythromycin, sulfonamide, and tetracycline, in samples of shrimp sourced from aquaculture operations in Southeast Asia [4,5].

Furthermore, several studies have indicated that shrimp is a carrier of antibiotic-resistant bacteria [6,7]. Antibiotic resistance genes present in multidrug-resistant (MDR) bacteria within shrimp can disseminate through horizontal gene transfer, potentially reaching pathogenic bacteria [8]. If a person becomes infected with these pathogenic bacteria, antibiotic treatments could be less effective. Consuming shrimp contaminated with MDR bacteria may facilitate the transfer of antibiotic resistance determinants between MDR and commensal bacteria. Elevated transfer rates of antibiotic resistance may potentially give rise to new antibiotic-resistant bacteria through the exchange of resistant genes within the human gut. This could disrupt the balance of human intestinal microbiota, posing a potential health risk [9]. Our objective in this study was to analyze antimicrobial resistance and virulence traits in MDR *E. coli* isolates from imported shrimp using whole-genome sequences (WGSs).

## 2. Materials and Methods

### 2.1. Isolation of E. coli from Imported Shrimp

A total of 330 frozen, imported shrimp were thawed and incubated with Luria broth (Thermo Fisher Scientific, Waltham, MA, USA) overnight. Next, the sample was streaked onto a MacConkey agar plate (Thermo Fisher Scientific), and the identity of an *E. coli* colony was confirmed using the Vitek GNI+ card (bioMerieux, Durham, NC, USA) and fatty acid methyl ester analysis (MIDI, Newark, DE, USA).

### 2.2. Phenotypic Analysis of Antibiotic Resistance

We employed the disk diffusion assay to assess the phenotypic antibiotic susceptibility of *E. coli* isolates [10]. It included antibiotic disks from diverse classes, such as aminoglycosides, beta-lactams, chloramphenicol, quinolones, fosfomycin, tetracycline, trimethoprim/sulfamethoxazole, and polymyxin B. Antimicrobial susceptibilities were assessed based on the criteria established by the Clinical and Laboratory Standards Institute [11].

### 2.3. Whole-Genome Sequencing

Genomic DNA were extracted from an overnight culture of 14 MDR *E. coli* using the DNeasy Blood and Tissue Kit (Qiagen, Valencia, CA, USA). DNA were fragmented using g-TUBE (Covaris, Woburn, MA, USA) to approximately 6.0 kb. We constructed libraries using the SMRTbell Express Template Prep Kit v2.0 (Pacific Biosciences, Menlo Park, CA, USA). SMRT cells of the libraries were sequenced using the PacBio Sequel II platform (Pacific Biosciences). Long-read sequencing data were assembled with the SMRT Link v10.0 Microbial Assembly application (Pacific Biosciences). Detailed information of the WGSs was reported by Alotaibi et al. [12] and sequencing data are available under the National Center for Biotechnology Information’s (NCBI) BioProject number PRJNA802087.

### 2.4. Identification of MLST, Serotype, Virulence, Antibiotic Resistance, and Mobile Elements

We determined multilocus sequence types (MLSTs) and serotypes of the assembled genomes by using MLST v.2.0. (https://cge.food.dtu.dk/services/MLST/ (accessed on 24 November 2023)), cgMLST v.1.2. (https://cge.food.dtu.dk/services/cgMLSTFinder/ (accessed on 5 May 2024)), and SerotypeFinder v.2.0 (https://cge.food.dtu.dk/services/SerotypeFinder/ (accessed on 25 November 2023)), available on the Center for Genomic Epidemiology (CGE) server [13]. Average nucleotide identity (ANI) values were determined by using Integrated Prokaryotes Genome and pangenome Analysis (IPGA) v.1.09 (https://nmdc.cn/ipga/ (accessed on 13 November 2023)) [14]. Antimicrobial resistance genes and point mutations were identified by using multiple databases, including the Comprehensive Antibiotic Resistance Database (CARD), the Bacterial and Viral Bioinformatics Resource Center (BV-BRC) (https://www.bv-brc.org/ (accessed on 16 January 2024)), the Bacterial Antimicrobial Resistance Reference Gene Database hosted by the NCBI as part of the National Database of Antibiotic Resistant Organisms (NDARO) (https://www.ncbi.nlm.nih.gov/pathogens/isolates/ (accessed on 20 October 2023)), and ResFinderPlus v.4.5.0. (http://genepi.food.dtu.dk/resfinder (accessed on 1 November 2023)) [15,16,17]. Mobile elements were identified by using Mobile Element Finder v.1.0.3 (https://cge.food.dtu.dk/services/MobileElementFinder/ (accessed on 27 November 2023)) on the CGE and mobileOG-db v1.1.3 of Proksee (https://proksee.ca/ (accessed on 28 November 2023)) [18,19]. Plasmid mobility and plasmid replicon were identified by using VRprofile2 (https://tool2-mml.sjtu.edu.cn/VRprofile/ (accessed on 3 December 2023)) [20]. Virulence factors were identified based on the datasets of the Virulence Factors of Pathogenic Bacteria Database (VFDB), the PATRIC curated virulence database (PATRIC_VF), and the Victors database [21,22,23].

### 2.5. Pangenome Analysis

We conducted pangenome analysis using CGView (https://server.gview.ca/ (accessed on 13 October 2023)) [24]. On the CGView server, Basic Local Alignment Search Tool (BLAST) analysis was performed using GenBank files of plasmid sequences, with an E value < 1 × 10^−10^, an alignment length cutoff value of 100, and a percent identity cutoff value of 80. The schematic maps of the complete genome structure of the plasmids were generated by using CGView.

## 3. Results

### 3.1. Antibiotic Resistance Profiles

Antibiotic resistance profiles of different *E. coli* isolates revealed a diverse landscape of susceptibility and resistance patterns across various antimicrobial agents (Figure 1). Notably, there was variability in resistance to a range of antibiotics among the isolates. For instance, isolates EC0002, EC110, EC119, EC120, EC123, EC126, EC331, EC335, EC338, EC339, EC1110, and EC2110 consistently exhibited resistance to gentamicin, streptomycin, ampicillin, chloramphenicol, ciprofloxacin, nalidixic acid, tetracycline, and trimethoprim/sulfamethoxazole. However, resistance to kanamycin and cefepime varied across these isolates, with some showing susceptibility or intermediate resistance.

### 3.2. MLST, cgMLST, Serotype, and ANI Analysis

The MLST profiles of diverse *E. coli* isolates, determined by specific gene alleles, revealed both shared characteristics and unique distinctions in their genetic composition (Table 1). Notably, isolates EC0002, EC110, EC119, EC120, EC123, EC126, EC331, EC335, EC338, EC339, EC1110, and EC2110 exhibited identical alleles across all seven MLST genes, resulting in the assignment of ST93. In contrast, isolates EC15 and ECV01 displayed distinctive alleles for several MLST genes, leading to the assignment of distinct sequence types (ST1148 and ST1196, respectively). The serotype profiles of *E. coli* isolates unveiled a predominant pattern across the strains (Table 1). Most of the isolates consistently exhibited O7 and H4 antigen types. In contrast, EC15 and ECV01 presented distinct serotypes with O163 and H7, and O22 and H28 antigens, respectively. To assess genomic relatedness and delineate distinctions among the *E. coli* isolates, we computed pairwise ANI for all possible genome pairs (Figure 2). The *E. coli* isolates were categorized into two groups, with EC15 and ECV01 forming a distinct, independent group. The ANI values across the bacterial isolates ranged from 98.33 to 100.00%.

### 3.3. Pangenome Analysis

For a more in-depth exploration of variations in the complete set of genes among *E. coli* strains, we conducted pangenome analysis utilizing the CGView server and its genome visualization tool. *E. coli* ATCC 9738 served as the reference genome, and the complete genomes of all isolates were included to generate the circular representation (Figure 3). The pangenome comparison results indicate that the genomes of the 12 isolates, excluding EC15 and ECV01, exhibited remarkably high similarity. Bacteriophage gene clusters, including portal protein, head-to-tail joining protein, lambda head decoration protein, major capsid protein, tail assembly protein, tail fiber protein, tail length tape measure protein, minor tail protein, lysin, DNA packaging protein, terminase, methyltransferase, and exonuclease, along with toxin–antitoxin system proteins and Hok/gef cell toxic protein were specifically present in the 2263–2310 kb region of EC15 (Appendix A). Conversely, flagella gene clusters, encompassing cytoplasmic chaperone, motor, export apparatus, hook-filament junction, rod, ring, and flagellar type III secretion system components, were exclusively identified in the 3673–3711 kb region of ECV01 (Appendix A).

**Figure 3 foods-13-01766-f003:**
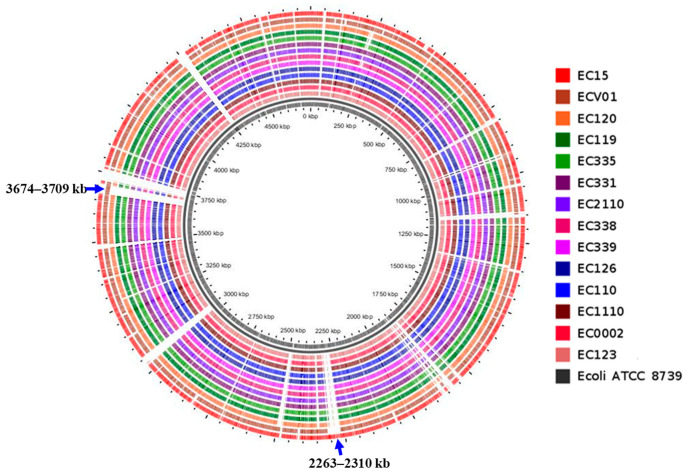
Pangenome mapping of *E. coli* genomes. The pangenome was constructed using CGView, involving the incorporation of distinctive regions into the reference genome (*E. coli* ATCC 8739). Gaps in the map signify regions where genes are absent in the reference genome but are present in other genomes. Pangenome analysis revealed a high degree of similarity among most plasmids. However, a small number contained additional DNA sequences incorporating specific genes (Figure 4A,B). We observed notable differences in the sequences of the pEC15-5 and pEC331-2 plasmids compared to others (Figure 4A,B). Specifically, pEC15-5 carried DNA replication and transfer genes, including *rop*, *mobC*, *mbeB*, and *mbeD*, while pEC331-2 contained tetracycline-resistant genes, *tetA* and *tetR*. Additionally, all plasmids shared the presence of aminoglycoside- and sulfonamide-resistant genes, including *aph(6)-Id*, *aph(3″)-Ib*, and *sul2* genes.

### 3.4. Antimicrobial Resistance and Virulence Genes in Chromosomes

Aminoglycoside, chloramphenicol, quaternary ammonium compound, and trimethoprim/sulfamethoxazole resistance genes, including *aac(3)-II,III,IV,VI,VIII,IX,X*, *aac(3)-IIa*, *aac(3)-IId*, *aadA2*, *aph(3″)-Ib*, *aph(6)-Id*, *catA2*, *qacE*, *dfrA12*, *sul1*, and *sul2,* were universally detected in all isolates, with the exception of EC15 and ECV01 (Figure 5). Additionally, ECV01 exclusively harbored *aadA1*, *cmlA1*, and *sul3* genes. Mutation patterns in key antibiotic resistance genes varied among isolates, with unique patterns for colistin (*pmrB*_Y358N in EC15 and ECV01) and quinolones (*gyrA*_D87N and *gyrA*_S83L shared; *parC*_E84K in all except EC15 and ECV01; *parC*_E84G exclusive to EC15; *parC*_S80I in EC15 and ECV01). The fosfomycin resistance-associated *cyaA*_S352T was absent in EC15 and ECV01, while *glpT*_E448K was universal, and *uhpT*_E350Q was exclusive to ECV01.

*E. coli* isolates exhibited varying numbers of genomic islands (GIs) and prophages. A clear pattern linking the number of GIs/prophages to the carriage of antibiotic resistance and virulence genes was not evident in Table 2. Isolate EC0002 contained 18 GIs and 9 prophages, hosting antibiotic resistance genes (*aac(3)-Iid*, *aadA2*, *blaTEM-1B*, *dfrA12*, *qacE*, and *sul1*), alongside virulence genes (*csgB*, *csgD-G*, *espL1*, *espX1*, *espX4*, *fimB*, *fimE*, *PAAR*, *rfaD*, *rhs*, *tssI*, and *vgrG*). In contrast, EC15, featuring 17 GIs and 7 prophages, displayed a repertoire of virulence genes such as *acrB*, *cap8E*, *csgA-G*, *espL1*, *espX4*, *fimA-C*, *fimE*, *fimI*, *hsiB1*, *hsiC1*, *spaP-Q*, and *vipA-B* but lacked antibiotic resistance genes. ECV01, characterized by 17 GIs and 5 prophages, exhibited a notable abundance of virulence genes without antibiotic resistance genes. Remarkably, *rfaD* was identified in EC0002, EC120, EC123, EC335, and ECV01, known for their association with enhanced biofilm formation and resistance to host defenses. Mobile element analysis showed commonalities (IS*609*, ISEc1, and MITEEc1) in all isolates except EC15 and ECV01 (Table 3). EC15 had IS*3*, IS*609*, ISEc1, ISSen1, and MITEEc1, while ECV01 had IS*609*, IS*621*, ISEc1, ISKpn8, ISSen1, and MITEEc1 exclusively.

A set of adhesion/invasion genes (*aslA*, *csgB*, *csgD-G*, *cusC*, *ecpA-E*, *ecpR*, *fdeC*, *fimA-I*, *flgC*, *flgG-H*, *fliG*, *fliM*, *fliP-Q*, *focA*, *hcpA*, *pitB*, *ppdD*, *serA*, *ydiV*, and *yijP*) was consistent across the chromosomes of all isolates (Figure 6). The intriguing absence of the flagellin (*fliC*) gene in all isolates except for EC15 and ECV01 added complexity to the observed gene profile variations among these isolates. Biofilm-related genes had consistent patterns, with *gatC*, *gatZ*, *kbaY*, and *luxS* present in all but *pgaA* absent in EC15 and ECV01. Capsule gene analysis revealed variable prevalence, with *etk*, *gfcA*, *gfcC-E*, and *ymcC* consistently present; however, *gfcB* varied, being absent in EC15 and ECV01. Iron acquisition genes *entA-F*, *entS*, *fepA-D*, *fepG*, *fes*, and *fur* were uniformly present, indicating shared iron acquisition capacity.

The prevalence of toxin genes varied, with the outer membrane efflux lipoprotein (*ibeB*) and outer membrane protein A (*ompA*) genes universally present. However, capsule polysaccharide synthesis genes were notably absent in EC15 and ECV01. Most genes associated with the type II (*gspC-M*) and III (*espL*, *espX*, and *map*) secretion systems were uniformly present, but EC15 and ECV01 lacked *espL4*. Genes related to the type I secretion system (*nuoG*) and general autotransporter pathway (*gatB*) genes were consistently present across all isolates.

### 3.5. Antimicrobial Resistance and Virulence Genes in Plasmids

Plasmids, including pEC0002-1, pEC110-1, pEC119-1, pEC120-1, pEC123-1, pEC126-1, pEC331-1, pEC335-1, pEC338-1, pEC339-1, pEC1110-1, and pEC2110-1, consistently carried resistance genes like *catA2*, *folA*, *tetA*, and *tetC* (Table 4). Conversely, plasmids such as pEC15-3, pEC15-4, and pEC15-5 appeared devoid of discernible antimicrobial resistance genes. Notably, pECV01-2 emerged as a comprehensive reservoir of antimicrobial resistance genes, featuring *aadA1*, *aadA2*, *ant(3″)-Ia*, *blaTEM-1B*, *cmlA1*, *dfrA12*, *mefB*, *qacE*, *sul3*, *tetA*, *tetR*, and *vgaC*. EC15 and ECV01 isolates exclusively possessed the streptogramin A resistance gene *vgaC*, distinguishing them from the remaining isolates.

Plasmids such as pEC15-1 and pEC15-2 showed an array of virulence genes, including *caf1A*, *caf1M*, *faeA*, *faeC-I*, *finP*, *ipaH*, and *virB* (Table 4); however, several plasmids exhibited no discernible virulence genes. Interestingly, pEC15-3 and pECV01-2 stood out as they possessed the virulence regulon transcriptional activator (*virB*) gene. The analysis of plasmid replicon types across the diverse set of *E. coli* plasmids revealed distinct patterns in their replication machinery (Table 4). Plasmids exceeding 123 kb in size exhibited a consistent IncFIB replicon type. Significantly, pEC15-2, pEC15-3, and pECV01-2 distinguished themselves by featuring IncI1-I, IncY, and IncR replicon types, setting them apart from the predominant IncFIB-dominated plasmids.

### 3.6. Mobile Elements in Plasmids

Integration/excision genes (*insO1*, *intA*, and *tnpA*) were present in all *E. coli* strains with plasmids of 123 kb in size, except for EC15 and ECV01 (Table 5). Notably, pECV01-2 showed an extensive set of integration/excision genes (*int*, *intM*, *ltrA*, *tnp*, *tnp1*, *tnpA*, and *tnpR*), while pEC15-3 uniquely carried *ref* and *rdgC*. Plasmids such as pEC15-1, pEC15-2, and pEC15-3 exhibited unique phage genes, including *ybiI_1*, *exc*, and *nusG*, and an extensive array associated with phage functions. Conversely, other plasmids shared common phage genes *cor* and *smc*. Plasmids carrying the type I restriction–modification system gene (*hsdR*) exhibited commonalities in their defense mechanisms.

In contrast, pEC15-2 and pEC15-3 carried antirestriction protein (*ardA*) and toxin-antitoxin system gene (*ccdA-B*), respectively. Plasmids such as pEC15-1 were equipped with an extensive set of transfer genes, including *finO*, *psiB*, *traA-E*, *traG-H*, *traK*, *traL-N*, *traT-U*, *traW-X*, *trbD*, and *trbI*. Conversely, pEC15-2 featured a different repertoire, with mobilization genes such as *mobB*, *nikB*, *pilI*, *pilK-U*, *psiB*, *sogL*, *traA*, *traC-R*, *traT-W*, *traY*, *trbA-C*, and *yggA*. Notably, pEC15-5 possessed *mbeB*, *mbeD*, and *mobC*, further contributing to its potential for efficient plasmid transfer. pEC15-2 featured a different set of insertion sequences, including ISEc25, ISEc37, ISEc43, Tn*2*, TnpA_Tn*3*, TnpA_TnAs1, TnpR_Tn*3*, and TnpR_TnEc1. Plasmids like pEC331-2 exhibited specific insertion sequences such as IS15DI and TnpA_TnAs1, contributing to the genetic diversity among these elements.

We constructed circular diagrams to facilitate the comparison of plasmid backbones. The EC15 isolate exhibited a spectrum of plasmids, ranging from the large pEC15-1 (144 kb, IncFIB) to the compact pEC15-5 (5 kb, IncFIB) (Appendix A). The comparison of gene profiles between two *E. coli* plasmids, pEC15 and pECV01, revealed both shared and distinct genetic elements within their compositions. Plasmid pEC15-1 carried virulence genes (*caf1A*, *caf1M*, and *virB*), fimbrial biosynthesis genes (*faeC-I*), plasmid replication genes (*repA* and *repB*), transfer genes (*traC-G*, *traK*, *traM-N*, *traR*, and *traU-X*), and various insertion sequences (IS*2*, IS*3*, IS*91*, and IS*629*). Plasmid pEC15-2, on the other hand, featured antibiotic resistance (*blaTEM-1B*, *tetA*, and *tetC*), mobilization (*mobC* and *nikB*), and transfer (*traA-B*, *traE-F*, *traH-I*, and *traM-Y*) genes. Plasmid pEC15-3 exhibited genes associated with phage activity (*kilA*), replication (*repA* and *repL*), and stability (*parA*). In contrast, plasmid pEC15-4 included *psiB*, while pEC15-5 harbored genes associated with mobilization (*mbeA*, *mbeD*, and *mobC*) and regulation (*rop*). pECV01-1 encompassed a range of genes involved in replication (*dnaE* and *dnaQ*), restriction–modification systems (*hsdR*), and transfer (*parB*, *recA*, and *repA*) (Appendix A). Notably, pECV01-2 had a comprehensive collection of antibiotic resistance genes (*blaTEM-1B*, *tetA*, *tetR*, and *cmlA1*) and virulence genes (*virB*), indicating a potential dual role.

## 4. Discussion

The MLST profiles provided insights into the genetic relatedness among the *E. coli* isolates, with most sharing a common sequence type (ST93). The shared alleles across seven MLST genes suggest potential clonality or a common evolutionary lineage among EC0002, EC110, EC119, EC120, EC123, EC126, EC331, EC335, EC338, EC339, EC1110, and EC2110. ST93 has been sporadically identified as both avian and human extra-intestinal pathogenic or diarrheagenic *E. coli* in various regions, encompassing humans, animals, and globally distributed food products [25,26]. Previous studies have reported its presence in beef, veal, pork, and poultry in Switzerland [27], a pig in Laos [28], a cat and retail food in China [29], broiler chickens in Brazil [30], and patients in Finland and Uruguay [31,32]. ECV01 exhibited a unique MLST type, ST1196, which has notably been associated with MDR *E. coli* in livestock, pets, hospitals, and wastewater across different countries, thus playing a pivotal role in the global transmission of antimicrobial resistance genes [33,34]. cgMLST also mirrors this pattern observed with MLST. Strains EC15 and ECV01 displayed distinct cgMLST types, 32199 and 69511, respectively, whereas all other isolates belonged to cgMLST type 39996.

The serotype analysis further supports the MLST findings, with most isolates exhibiting a consistent serological identity (O7:H4). Interestingly, a recent outbreak of food poisoning in Japan occurred among individuals who consumed seaweed [35]. The culprit behind the illness was identified as O7:H4 serotype *E. coli*. Like our strains of *E. coli*, this variant also exhibited resistance to β-lactam antibiotics. Unique serotypes of EC15 (O163:H7) and ECV01 (O22:H28) underscore the diversity within this bacterial population, suggesting distinct evolutionary trajectories. ANI analysis provided additional granularity to the genomic relationships among isolates, categorizing them into two groups. EC15 and ECV01 formed a distinct group, reinforcing their genetic distinctiveness. The high ANI values within each group signify genomic similarity, while the variations between groups highlight the genetic diversity in the *E. coli* population.

In aquaculture, the acknowledged prophylactic and therapeutic application of tetracycline has emerged as a significant contributor to the dissemination of tetracycline resistance genes in the environment [36]. Remarkably, these resistance genes were persistent in aquaculture settings, even in the absence of continual tetracycline use [37]. Notably, all *E. coli* isolates in our study were found to carry both *tetA* and *tetC* genes. Consistent with our findings, earlier research has highlighted the prevalence of tetracycline resistance genes among bacteria recovered from imported shrimps originating from Asian countries [38].

Moreover, investigations have indicated the dissemination of plasmids carrying tetracycline resistance determinants between aquaculture and human environments [39]. The presence of chloramphenicol residue is a common cause for the rejection of imported shrimp in the U.S. [40], indicative of the widespread use of banned antibiotics in shrimp-exporting countries [41]. Previous investigations into *E. coli* isolates from fish and shrimp in Vietnam have revealed a significant number of isolates resistant to chloramphenicol [42,43]. Elevated occurrences of chloramphenicol resistance genes have also been documented in *E. coli* isolates from fish, shrimp, and shellfish in China [44]. Ng et al. detected *E. coli* isolates containing *catA2* from aquaculture farms in Malaysia [45]. Notably, all *E. coli* isolates from imported shrimp were resistant to chloramphenicol and harbored the chloramphenicol resistance genes *catA2* and *cmlA1*.

Sulfonamide is extensively used in animal production systems, either as a standalone drug or in conjunction with diaminopyrimidines [46]. Notably, a heightened prevalence of sulfonamide resistance has been documented in enteric bacteria isolated from both livestock and humans [47,48]. Gram-negative bacteria exhibiting resistance to sulfonamides have been reported in shrimp farms in Vietnam [49]. This resistance is frequently linked to the acquisition of *sul2* [48]. The *sul2* gene has predominantly been identified on small plasmids, and this pattern extends to our *E. coli* isolates from imported shrimp, where *sul2* was also located on small plasmids. The identification of chloramphenicol (*catA2*), aminoglycoside- (*aph(3″)-Ib* and *aph(6)-Id*), tetracycline- (*tetA* and *tetC*), trimethoprim- (*folA*), and sulfonamide (*sul2*)-resistant genes within plasmids of the *E. coli* isolates from imported shrimp emphasizes their contribution to the dissemination of antibiotic resistance.

The comparative analysis of the chromosomes and plasmids of diverse *E. coli* isolates sheds light on the intricate interplay between intrinsic and extrachromosomal genetic elements, providing a comprehensive understanding of the factors influencing antibiotic resistance, virulence, and adaptability within this bacterial population. Examination of antimicrobial resistance genes in both chromosomes and plasmids reveals distinct patterns. While the chromosomes harbored a conserved set of resistance genes, including *aac(3)-II,III,IV,VI,VIII,IX,X*, *aac(3)-IIa*, *aac(3)-IId*, *aadA2*, *aph(3″)-Ib*, *aph(6)-Id*, *catA2*, *pmrE*, *qacE*, *dfrA12*, *sul1*, and *sul2*, the plasmids exhibited variability in resistance profiles. Plasmids such as pEC15-3, pEC15-4, pEC15-5, and pEC119-3 did not contain discernible antimicrobial resistance genes. In contrast, plasmids like pECV01-2 emerged as comprehensive reservoirs of antimicrobial resistance genes, hosting a diverse array that includes *aadA1*, *aadA2*, *ant(3″)-Ia*, *blaTEM-1B*, *cmlA1*, *dfrA12*, *mefB*, *qacE*, *sul3*, *tetA*, *tetR*, and *vgaC*. This suggests a potential reliance on plasmid determinants for antibiotic resistance in this isolate.

Antibiotic resistance ensures bacterial, and consequently plasmid, survival when the relevant resistance gene is carried on the plasmid [50]. However, the transfer of the gene to the chromosome preserves the selective advantage of resistance while alleviating the fitness cost associated with replicating large plasmids. Among the examined isolates, twelve were found to carry the genes *aadA2*, *qacE*, and *dfrA12*; notably, only one isolate was observed to harbor these genes within a plasmid, while the rest housed them in the chromosome. This observation aligns with the proposed evolutionary pressure for plasmid-borne antimicrobial resistance genes to move to the chromosome. On the other hand, certain determinants such as *catA2*, *tetC*, *tetR*, and *vgaC* genes were exclusively found on plasmids, highlighting the diversity in the genetic distribution of resistance determinants among bacterial populations.

Antibiotic resistance genes associated with efflux pumps are predominantly encoded on chromosomes, with *E. coli* efflux pump genes like *acrAB-tolC*, *acrAD-tolC*, *acrEF-tolC*, *mdtABC-tolC*, and *mdtEF-tolC* often integrated into the core genome, serving diverse cellular functions [51,52]. Various plasmid-borne efflux pump genes, including *tetA*, *tetX3*, *tetX4*, *tetX5*, *qepA*, *oqxAB*, *acrR*, *tmexCD1-toprJ1*, *sugE*, *qacA/B*, *qacA*, *qacG*, and *qacH*, have been identified in both Gram-negative and Gram-positive bacteria, contributing to antibiotic resistance [53,54,55]. However, several efflux pump genes, such as acid resistance transcriptional activators (*gadE*, *gadW*, and *gadX*), resistance-nodulation-division-type efflux pumps (*mdtE* and *mdtF*), and a metabolite, H1 symporter (MHS) family major facilitator superfamily transporter (*MNZ41_23075*), were found in the plasmid of the EC119 isolate. To our knowledge, there has been no reported instance of bacteria harboring such a diverse combination of efflux pump genes within a plasmid. The unique presence of this combination in the plasmid of EC119 underscores the need for further research that would enhance our understanding of its implications for antibiotic resistance.

Our analysis of mobile genetic elements in plasmids illuminates their remarkable potential for movement and adaptation within bacterial populations. The presence of integration/excision genes (*insO1*, *intA*, and *tnpA*) in the majority of plasmids suggests their capacity for mobilization. These genes function akin to molecular cut-and-paste tools, facilitating the integration of plasmid DNA into the host chromosome, potentially disseminating antimicrobial resistance and virulence genes within the bacterium [56]. Notably, plasmid pECV01-2 harbors an extensive array of integration/excision genes (*int*, *intM*, *ltrA*, *tnp*, *tnp1*, *tnpA*, and *tnpR*), indicating a potentially heightened role in its own mobilization compared to other plasmids. This dynamic interaction suggests active participation of the plasmid in seeking opportunities to spread its genetic payload. Distinct sets of transfer genes across plasmids imply diverse dissemination strategies. Plasmid pEC15-1 appears well equipped with a comprehensive set of conjugation-related genes like *finO*, *traA-E*, and *traG-H*, facilitating the direct transfer of plasmid DNA between bacterial cells. Furthermore, the presence of *mbeB*, *mbeD*, and *mobC* in pEC15-5 suggests their involvement in plasmid mobilization, indicating a sophisticated network of transfer mechanisms. These findings underscore the adaptability of plasmids, employing diverse strategies for successful dissemination within bacterial communities. Beyond transfer, our analysis reveals mechanisms ensuring plasmid stability within host bacteria. Notably, pEC15-3 carries a toxin–antitoxin system gene (*ccdA-B*), serving as a “molecular suicide switch” where a loss of the plasmid triggers host cell inactivation by the toxin [57]. This strategy confers a selective advantage to plasmid-carrying cells, promoting their survival and proliferation, thus ensuring plasmid persistence alongside host lineages. Interestingly, pEC15-2 encodes an antirestriction protein (*ardA*), potentially countering host or other mobile genetic element-encoded restriction enzymes [58], safeguarding plasmid integrity against degradation.

Analysis of virulence gene categories, spanning adhesin, invasion, biofilm formation, capsule, chemotaxis, iron acquisition, toxin, and secretion system genes, revealed the intricate landscape of genetic determinants within chromosomes. In the EC15 isolate, genes associated with adherence, invasion, and capsule were housed in plasmids. Unique to EC15’s plasmids were *ipaH* and *virB* genes, implicated in triggering cell death and modulating host inflammatory signals during bacterial infection [59]. The EC15 isolate also harbored K88 fimbrial biosynthesis genes on plasmids, known as major colonization factors in some enterotoxigenic *E. coli* strains associated with porcine neonatal and postweaning diarrhea [60]. The concurrent presence of type 1C and K88 fimbrial biosynthesis genes on chromosomes (*fimA-I*) and plasmids (*faeA* and *faeC-I*) suggests a collaborative role in adherence, a crucial aspect of bacterial tissue colonization. The presence of distinct capsule genes on both the chromosome (*etk*, *gfcA-E*, and *ymcC*) and plasmid (*caf1A*, *caf1M*) in the EC15 isolate implies a sophisticated immune evasion strategy, potentially surpassing that of other *E. coli* strains. The findings that we obtained from comprehensive whole-genome analyses of the *E. coli* isolates offer insights into the molecular basis of MDR and virulence in *E. coli*.

## 5. Conclusions

In summary, comprehensive analysis of the genetic elements in the chromosomes and plasmids of diverse *E. coli* isolates provides a holistic understanding of the factors influencing antibiotic resistance, pathogenicity, and adaptability. The intricate interplay between mobile elements, genomic islands, and various functional gene categories contributes to the dynamic nature of the *E. coli* population and has implications for public health, emphasizing the need for continued surveillance and research into the evolution of these important bacterial pathogens.

## Figures and Tables

**Figure 1 foods-13-01766-f001:**
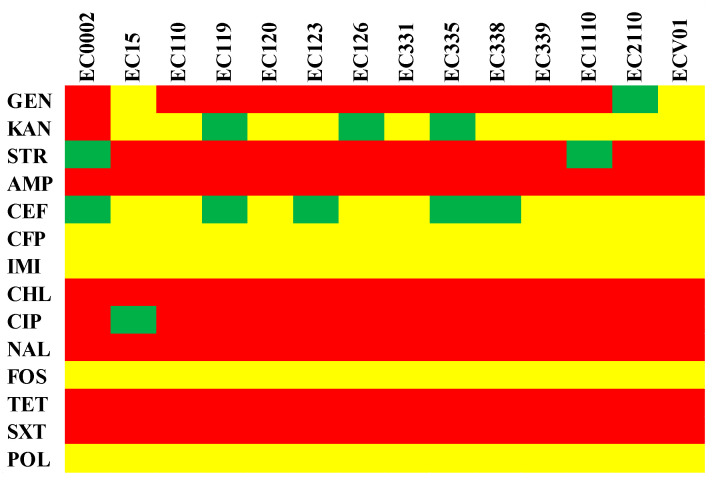
Phenotypic antibiotic resistance profile of *E. coli* isolates. **GEN**: gentamicin, **KAN**: kanamycin, **STR**: streptomycin, **AMP**: ampicillin, **CEF**: cefepime, **CFP**: cefoperazone, **IMI**: imipenem, **CHL**: chloramphenicol, **CIP**: ciprofloxacin, **NAL**: nalidixic acid, **FOS**: fosfomycin, **TET**: tetracycline, **SXT**: trimethoprim/sulfamethoxazole, and **POL**: polymyxin B. The figure shows antimicrobial susceptibility as resistant (red), intermediate (green), or susceptible (yellow).

**Figure 2 foods-13-01766-f002:**
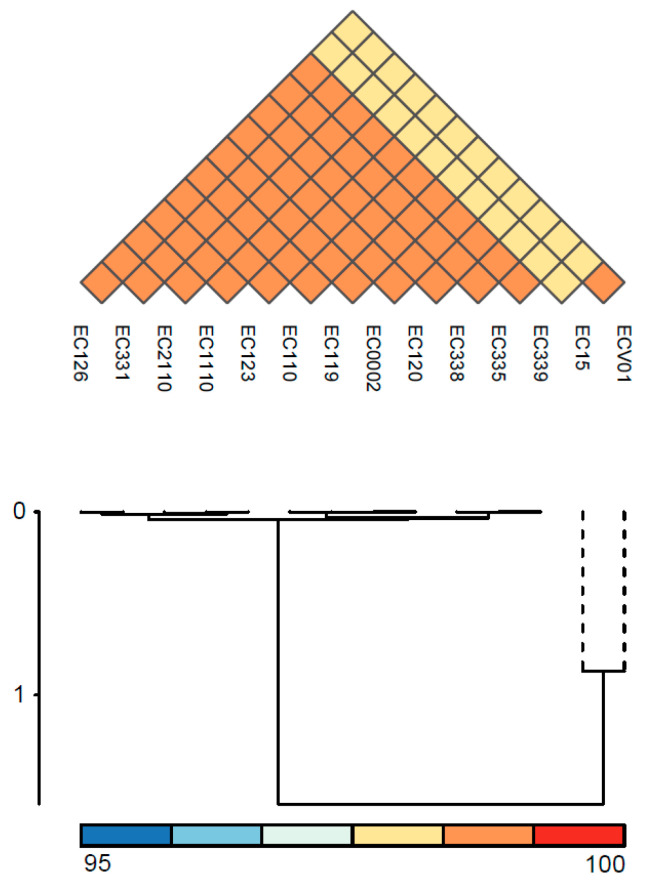
ANI analysis of *E. coli* isolates. The color indicates the value of ANI; the value range is 95–100, with the color turning from blue to red.

**Figure 4 foods-13-01766-f004:**
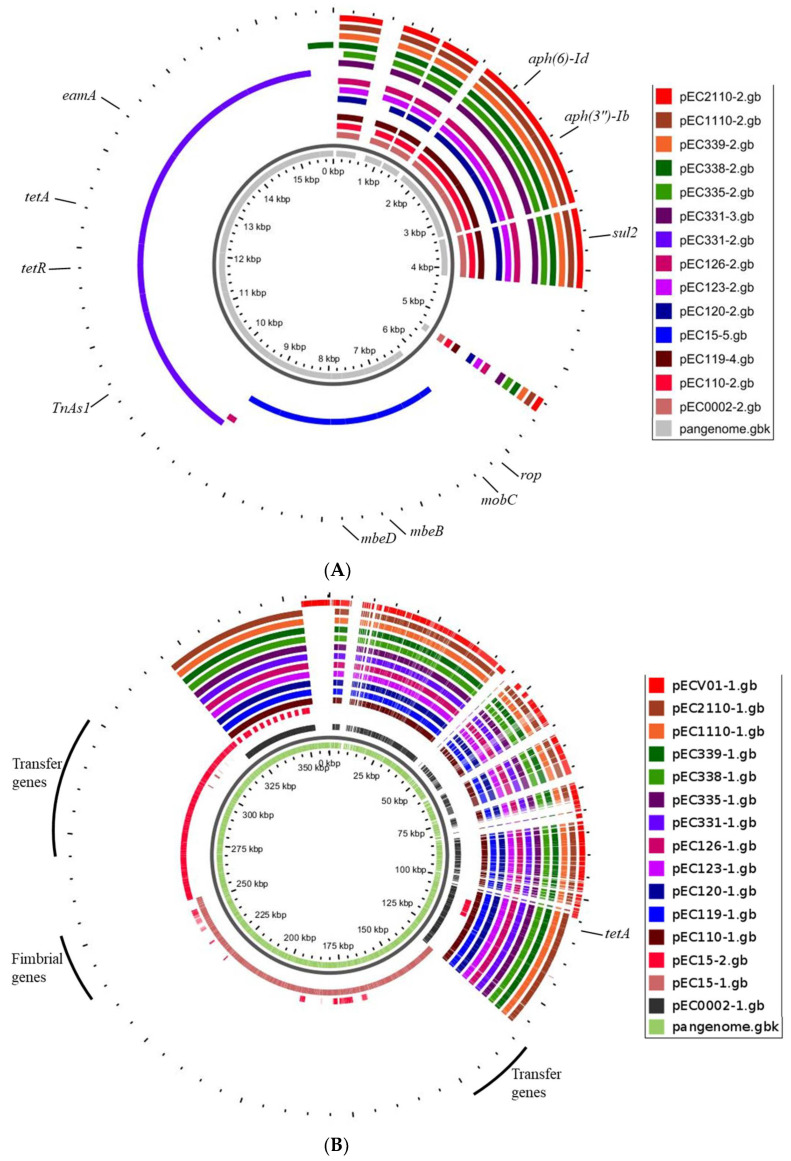
Pangenome mapping of plasmids within the size range of 5.2–6.5 kb (**A**) and 107–124 kb (**B**) across *E. coli* isolates. The innermost circle depicts the pangenome in gray, while the outer circles represent individual plasmids.

**Figure 5 foods-13-01766-f005:**
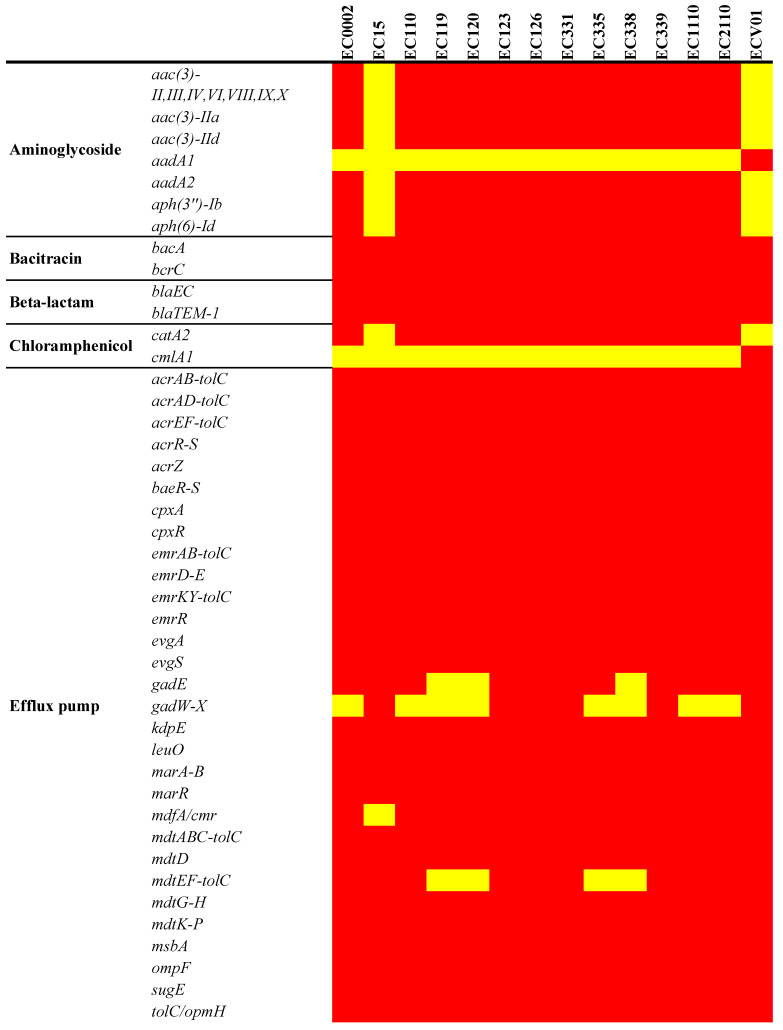
Antimicrobial resistance determinants in the *E. coli* chromosome. The figure shows the presence (red for antimicrobial resistance genes or green for point mutations) or absence (yellow) of antimicrobial resistance genes. * Quaternary ammonium compound.

**Figure 6 foods-13-01766-f006:**
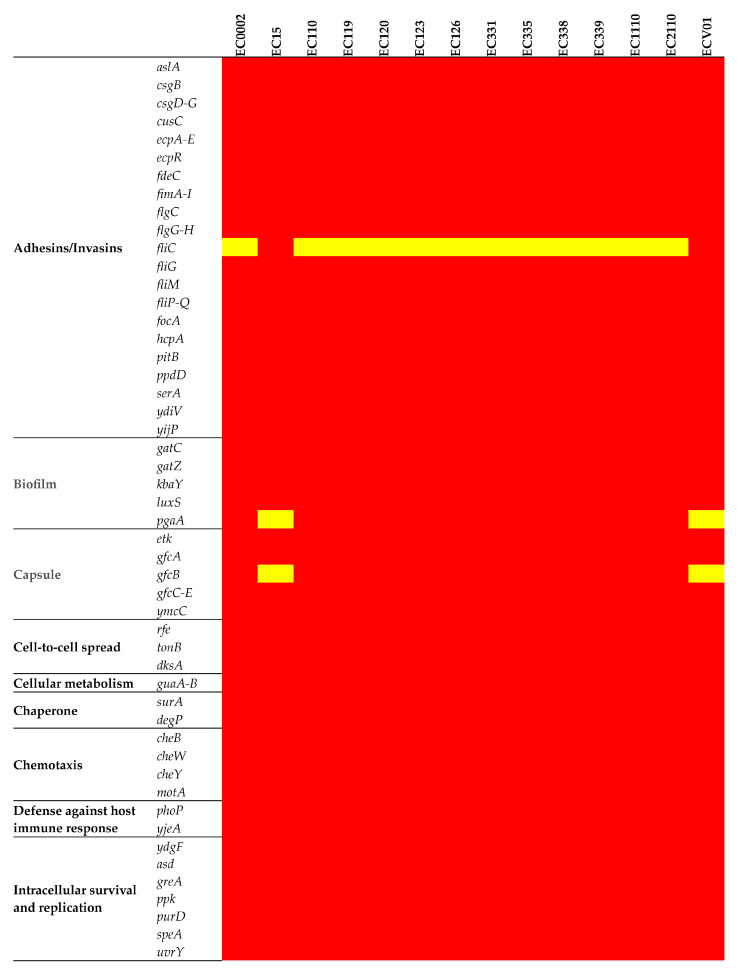
Virulence determinants in the *E. coli* chromosome. The figure shows the presence (red) or absence (yellow) of virulence genes.

**Table 1 foods-13-01766-t001:** Multilocus sequence and serotyping of *E. coli* isolates.

		EC0002	EC15	EC110	EC119	EC120	EC123	EC126	EC331	EC335	EC338	EC339	EC1110	EC2110	ECV01
**Multilocus sequence typing**	** *adk* **	6	6	6	6	6	6	6	6	6	6	6	6	6	6
	** *fumC* **	11	95	11	11	11	11	11	11	11	11	11	11	11	6
	** *gyrB* **	4	3	4	4	4	4	4	4	4	4	4	4	4	33
	** *icd* **	10	18	10	10	10	10	10	10	10	10	10	10	10	26
	** *mdh* **	7	11	7	7	7	7	7	7	7	7	7	7	7	11
	** *purA* **	8	7	8	8	8	8	8	8	8	8	8	8	8	8
	** *recA* **	6	14	6	6	6	6	6	6	6	6	6	6	6	2
	**ST**	**93**	**1148**	**93**	**93**	**93**	**93**	**93**	**93**	**93**	**93**	**93**	**93**	**93**	**1196**
**cgMLST**		39996	32199	39996	39996	39996	39996	39996	39996	39996	39996	39996	39996	39996	69511
**Serotyping**	**O type**	O7	O163	O7	O7	O7	O7	O7	O7	O7	O7	O7	O7	O7	O22
	**H type**	H4	H7	H4	H4	H4	H4	H4	H4	H4	H4	H4	H4	H4	H28

**Table 2 foods-13-01766-t002:** Antibiotic resistance and virulence genes in genomic islands (GIs) and prophages of the chromosome. **Bold**: identified from prophages.

	Number of GIs	Number of Prophages	Antibiotic Resistance Genes	Virulence Genes
**EC0002**	18	9	*aac(3)-Iid, aadA2, blaTEM-1B, dfrA12, qacE, sul1*	*csgB, csgD-G, espL1, espX1, espX4, fimB, fimE, PAAR, **rfaD**, rhs, tssI, vgrG*
**EC15**	17	7		*acrB, cap8E, csgA-G, espL1, espX4, fimA-C, fimE, fimI, hsiB1, hsiC1, spaP-Q, vipA-B*
**EC110**	17	6	*aac(3)-Iid, aadA2, blaTEM-1B, dfrA12, qacE, sul1*	*csgA-G, entD, espL1, espX1, espX4, fimA-I*
**EC119**	17	6	*aac(3)-Iid, aadA2, blaTEM-1B, dfrA12, qacE, sul1*	*csgA-G, entD, espL1, espX1, espX4, fimA-I, **gmhA, lpcA***
**EC120**	18	9	*aac(3)-Iid, aadA2, blaTEM-1B, dfrA12, qacE, sul1*	*csgB, csgD-G, espL1, espX1, espX4, fimB, fimE, PAAR, **rfaD**, rhs, tssI, vgrG*
**EC123**	17	9	*aac(3)-Iid, aadA2, blaTEM-1B, dfrA12, qacE, sul1*	*csgB, csgD-G, espL1, espX1, espX4, fimB, fimE, PAAR, **rfaD**, rhs, tssI, vgrG*
**EC126**	17	6	*aac(3)-Iid, aadA2, blaTEM-1B, dfrA12, qacE, sul1*	*csgA-G, entD, espL1, espX1, espX4, fimA-I*
**EC331**	16	9	*aac(3)-Iid, aadA2, blaTEM-1B, dfrA12, qacE, sul1*	*csgB, csgD-G, espL1, espX1, espX4, fimB, fimE, PAAR, rhs, tssI, vgrG*
**EC335**	18	9	*aac(3)-Iid, aadA2, blaTEM-1B, dfrA12, qacE, sul1*	*csgB, csgD-G, espL1, espX1, espX4, fimB, fimE, **gmhA, lpcA**, PAAR, **rfaD**, rhs, tssI, vgrG*
**EC338**	15	8	*aac(3)-Iid, aadA2, blaTEM-1B, dfrA12, qacE, sul1*	*csgB, csgD-G, espL1, espX1, espX4, fimB, fimE, PAAR, rhs, tssI, vgrG*
**EC339**	17	6	*aac(3)-Iid, aadA2, blaTEM-1B, dfrA12, qacE, sul1*	*csgA-G, entD, espL1, espX1, espX4, fimA-I, **gmhA, lpcA***
**EC1110**	17	7	*aac(3)-Iid, aadA2, blaTEM-1B, dfrA12, qacE, sul1*	*csgA-G, entD, espL1, espX1, espX4, fimA-I*
**EC2110**	16	7	*aac(3)-Iid, aadA2, blaTEM-1B, dfrA12, qacE, sul1*	*csgA-G, entD, espL1, espX1, espX4, fimA-I*
**ECV01**	17	5		*acrB, csgB, csgD-G, ecpA-E, ecpR, espL1, espX4, fdeC, **fimB, fimE,** galU, gmhA, hsiB1, hsiC1, lpcA, rfaD, spaP-Q, vipA-B*

**Table 3 foods-13-01766-t003:** Mobile elements of *E. coli* chromosome. **Bold**: common in all isolates. **Italics**: common in all isolates except EC15 and ECV01.

Strains	Mobile Element
**EC0002**	*IS3*, *IS26*, *IS30*, *IS421*, **IS609**, *ISCfr1*, **ISEc1**, *ISEc17*, *ISEc31*, *ISEcB1*, *ISKpn8*, **MITEEc1**, cn_5709_IS26
**EC15**	IS3, **IS609**, **ISEc1**, ISSen1, **MITEEc1**
**EC110**	cn_2244_ISEc1, cn_5709_IS26, *IS3*, *IS26*, *IS30*, *IS421*, **IS609**, *ISCfr1*, **ISEc1**, *ISEc17*, *ISEc31*, *ISEcB1*, *ISKpn8*, **MITEEc1**
**EC119**	cn_2244_ISEc1, cn_5709_IS26, *IS3*, *IS26*, *IS30*, *IS421*, **IS609**, *ISCfr1*, **ISEc1**, *ISEc17*, *ISEc31*, *ISEcB1*, *ISKpn8*, **MITEEc1**
**EC120**	cn_5709_IS26, *IS3*, *IS26*, *IS30*, *IS421*, **IS609**, *ISCfr1*, **ISEc1**, *ISEc17*, *ISEc31*, *ISEcB1*, *ISKpn8*, **MITEEc1**
**EC123**	cn_5709_IS26, *IS3*, *IS26*, *IS30*, *IS421*, **IS609**, *ISCfr1*, **ISEc1**, *ISEc17*, *ISEc31*, *ISEcB1*, *ISKpn8*, **MITEEc1**
**EC126**	cn_2244_ISEc1, cn_5709_IS26, *IS3*, *IS26*, *IS30*, *IS421*, **IS609**, *ISCfr1*, **ISEc1**, *ISEc17*, *ISEc31*, *ISEcB1*, *ISKpn8*, **MITEEc1**
**EC331**	cn_5709_IS26, *IS3*, *IS26*, *IS30*, *IS421*, **IS609**, *ISCfr1*, **ISEc1**, *ISEc17*, *ISEc31*, *ISEcB1*, *ISKpn8*, **MITEEc1**
**EC335**	cn_5709_IS26, *IS3*, *IS26*, *IS30*, *IS421*, **IS609**, *ISCfr1*, **ISEc1**, *ISEc17*, *ISEc31*, *ISEcB1*, *ISKpn8*, **MITEEc1**
**EC338**	cn_5709_IS26, *IS3*, *IS26*, *IS30*, *IS421*, **IS609**, *ISCfr1*, **ISEc1**, *ISEc17*, *ISEc31*, *ISEcB1*, *ISKpn8*, **MITEEc1**
**EC339**	cn_2244_ISEc1, cn_5709_IS26, *IS3*, *IS26*, *IS30*, *IS421*, **IS609**, *ISCfr1*, **ISEc1**, *ISEc17*, *ISEc31*, *ISEcB1*, *ISKpn8*, **MITEEc1**
**EC1110**	cn_2244_ISEc1, cn_5709_IS26, *IS3*, *IS26*, *IS30*, *IS421*, **IS609**, *ISCfr1*, **ISEc1**, *ISEc17*, *ISEc31*, *ISEcB1*, *ISKpn8*, **MITEEc1**
**EC2110**	cn_2244_ISEc1, cn_5709_IS26, *IS3*, *IS26*, *IS30*, *IS421*, **IS609**, *ISCfr1*, **ISEc1**, *ISEc17*, *ISEc31*, *ISEcB1*, *ISKpn8*, **MITEEc1**
**ECV01**	cn_11046_ISEc1, **IS609**, IS621, **ISEc1**, ISKpn8, ISSen1, **MITEEc1**

**Table 4 foods-13-01766-t004:** Antibiotic resistance and virulence genes, replicon types, and sizes of *E. coli* plasmids.

Strains	Antimicrobial Resistant Genes	Virulence Genes	Replicon Type	Mobility	Sizes (bp)
**pEC0002-1**	*catA2, folA, tetA, tetC*	None	IncFIB	Non-Mobilizable	123,909
**pEC0002-2**	*aph(3″)-Ib, aph(6)-Id, sul2*	None	None	Mobilizable	6222
**pEC15-1**	*vgaC*	*caf1A, caf1M, faeA, faeC-I, finP, ipaH, virB*	IncFIB, IncFII	Non-Mobilizable	144,323
**pEC15-2**	*blaTEM-1B, tetA, tetC*	*faeA, faeC-J, ipaH*	IncI1-I	Conjugative	106,546
**pEC15-3**	*None*	virB	IncY	Non-Mobilizable	97,943
**pEC15-4**	*None*	None	None	Non-Mobilizable	12,743
**pEC15-5**	*None*	None	None	Mobilizable	5284
**pEC110-1**	*catA2, folA, tetA, tetC*	None	IncFIB	Non-Mobilizable	123,909
**pEC110-2**	*aph(3″)-Ib, aph(6)-Id, sul2*	None	None	Mobilizable	6222
**pEC119-1**	*catA2, folA, tetA, tetC*	None	IncFIB	Non-Mobilizable	123,904
**pEC119-2**	*gadE, gadX, gadW, mdtE, mdtF*	*acrB, yhiE*	None	Non-Mobilizable	32,064
**pEC119-3**	*None*	None	None	Non-Mobilizable	15,716
**pEC119-4**	*aph(3″)-Ib, aph(6)-Id, sul2*	None	None	Mobilizable	6222
**pEC120-1**	*catA2, folA, tetA, tetC*	None	IncFIB	Non-Mobilizable	123,909
**pEC120-2**	*aph(3″)-Ib, aph(6)-Id, sul2*	None	None	Mobilizable	6222
**pEC123-1**	*catA2, folA, tetA, tetC*	None	IncFIB	Non-Mobilizable	123,909
**pEC123-2**	*aph(3″)-Ib, aph(6)-Id, sul2*	None	None	Mobilizable	6222
**pEC126-1**	*catA2, folA, tetA, tetC*	None	IncFIB	Non-Mobilizable	123,909
**pEC126-2**	*aph(3″)-Ib, aph(6)-Id, sul2*	None	None	Mobilizable	6222
**pEC331-1**	*catA2, folA, tetA, tetC*	None	IncFIB	Non-Mobilizable	123,912
**pEC331-2**	*tetA, tetR*	None	None	Non-Mobilizable	6458
**pEC331-3**	*aph(3″)-Ib, aph(6)-Id, sul2*	None	None	Mobilizable	6222
**pEC335-1**	*catA2, folA, tetA, tetC*	None	IncFIB	Non-Mobilizable	123,909
**pEC335-2**	*aph(3″)-Ib, aph(6)-Id, sul2*	None	None	Mobilizable	6222
**pEC338-1**	*catA2, folA, tetA, tetC*	None	IncFIB	Non-Mobilizable	123,909
**pEC338-2**	*aph(3″)-Ib, aph(6)-Id, sul2*	None	None	Mobilizable	6222
**pEC339-1**	*catA2, folA, tetA, tetC*	None	IncFIB	Non-Mobilizable	123,909
**pEC339-2**	*aph(3″)-Ib, aph(6)-Id, sul2*	None	None	Mobilizable	6222
**pEC1110-1**	*catA2, folA, tetA, tetC*	None	IncFIB	Non-Mobilizable	123,909
**pEC1110-2**	*aph(3″)-Ib, aph(6)-Id, sul2*	None	None	Mobilizable	6222
**pEC2110-1**	*catA2, folA, tetA, tetC*	None	IncFIB	Non-Mobilizable	123,909
**pEC2110-2**	*aph(3″)-Ib, aph(6)-Id, sul2*	None	None	Mobilizable	6222
**pECV01-1**	*folA*	None	IncFIB	Non-Mobilizable	107,661
**pECV01-2**	*aadA1, aadA2, ant(3″)-Ia, blaTEM-1B, cmlA1, dfrA12, mefB, qacE, sul3, tetA, tetR, vgaC*	*virB*	IncR	Mobilizable	44,489

**Table 5 foods-13-01766-t005:** Mobile elements of *E. coli* plasmids.

Strains	Integration/ Excision	Replication/ Recombination/Repair	Phage	Stability/ Transfer/ Defense	Transfer	Insertion Sequences
**pEC0002-1**	*insO1, intA, tnpA*	*dnaG, dnaQ_2, parB_3, polA, recA, rnhA, uvrB_2*	*cor*, *smc*	*hsdR*	*dut*	cn_3039_IS26, IS26, IS911
**pEC0002-2**				*exc1*		
**pEC15-1**	*insC2, insD1, insE3, insF3, insG, insK, intM, NGR_a00170, NGR_a00180, S4062, tnpB-C, tnpR*	*impA-C, parA-B, repA, repA2, ssb, yfhA*	*ybiI_1*	*ccdA-B, vagC, vapC, yubI*	*finO, psiB, traA-E, traG-H, traK, traL-N, traT-U, traW-X, trbD, trbI*	IS2, IS3, IS3411, IS4, IS629, IS91, ISCro1, ISEc12, ISEc17, ISEc20, ISEc25, ISEc27, ISEc37, ISEc43
**pEC15-2**	*bin3_2, intM, S4062, tnpA, tnpC, tnpR*	*impB-C, parA, parM, repA, ssb, yfhA, yhgA*	*exc, nusG*	*ardA, stbD, yubI*	*mobC, nikB, pilI, pilK-U, psiB, sogL, traA-B, traE-F, traH-I, traM-Y, trbA-C, yggA*	ISEc25, ISEc37, ISEc43, Tn2, TnpA_Tn3, TnpA_TnAs1, TnpR_Tn3, TnpR_TnEc1
**pEC15-3**	*ref, rdgC*	*dnaB, holE_2, parA-B, repA, repL, ssb, umuD_5*	*ant, bof, cre, darA-B, ea22, gp6, gp21, gp23, kilA, lpa, pacA-B, R, S, sit, tfaQ_1*	*dam, MOD, phd, res, tec*		
**pEC15-4**		*yfhA, yhgA*		*ardA*	*psiB, yggA*	
**pEC15-5**		*rop*			*mbeA, mbeD, mobC*	
**pEC110-1**	*insO1, intA, tnpA*	*dnaG, dnaE, dnaQ_2, parB_3, polA, recA, repA, rnhA, uvrB_2*	*cor*, *smc*	*hsdR*	*dut*	cn_2812_IS26, IS26, IS911
**pEC110-2**				*exc1*		
**pEC119-1**	*insO1, intA, tnpA*	*dnaE, dnaG, dnaQ_2, parB_3, polA, recA, repA, rnhA, uvrB_2*	*cor*, *smc*	*hsdR*	*dut*	cn_3034_IS26, IS15DI, IS26, IS911
**pEC119-2**	*slp, insA8, insB*					IS1R, ISKpn14
**pEC119-3**	*tnpA*	*dnaE, polA, recA*				IS26
**pEC119-4**				*exc1*		
**pEC120-1**	*insO1, intA, tnpA*	*dnaE, dnaG, dnaQ_2, parB_3, polA, recA, repA, rnhA, uvrB_2*	*cor*, *smc*	*hsdR*	*dut*	cn_3039_IS26, IS26, IS911
**pEC120-2**				*exc1*		
**pEC123-1**	*insO1, intA, tnpA*	*dnaE, dnaG, dnaQ_2, parB_3, polA, recA, repA, rnhA, uvrB_2*	*cor*, *smc*	*hsdR*	*dut*	cn_3039_IS26, IS26, IS911
**pEC123-2**				*exc1*		
**pEC126-1**	*insO1, intA, tnpA*	*dnaE, dnaG, dnaQ_2, parB_3, polA, recA, repA, rnhA, uvrB_2*	*cor*, *smc*	*hsdR*	*dut*	cn_3039_IS26, IS26, IS911
**pEC126-2**				*exc1*		
**pEC331-1**	*insO1, intA, tnpA*	*dnaE, dnaG, dnaQ_2, parB_3, polA, recA, repA, rnhA, uvrB_2*	*cor*, *smc*	*hsdR*	*dut*	cn_3039_IS26, IS26, IS911
**pEC331-2**	*tnpA*					IS15DI, TnpA_TnAs1
**pEC331-3**				*exc1*		
**pEC335-1**	*insO1, intA, tnpA*	*dnaE, dnaG, dnaQ_2, parB_3, polA, recA, repA, rnhA, uvrB_2*	*cor*, *smc*	*hsdR*	*dut*	cn_3039_IS26, IS26, IS911
**pEC335-2**				*exc1*		
**pEC338-1**	*insO1, intA, tnpA*	*dnaE, dnaG, dnaQ_2, parB_3, polA, recA, repA, rnhA, uvrB_2*	*cor*, *smc*	*hsdR*	*dut*	cn_3039_IS26, IS26, IS911
**pEC338-2**				*exc1*		
**pEC339-1**	*insO1, intA, tnpA*	*dnaE, dnaG, dnaQ_2, parB_3, polA, recA, repA, rnhA, uvrB_2*	*cor*, *smc*	*hsdR*	*dut*	cn_3039_IS26, IS26, IS911
**pEC339-2**				*exc1*		
**pEC1110-1**	*insO1, intA, tnpA*	*dnaE, dnaG, dnaQ_2, parB_3, polA, recA, repA, rnhA, uvrB_2*	*cor*, *smc*	*hsdR*	*dut*	cn_3039_IS26, IS26, IS911
**pEC1110-2**				*exc1*		
**pEC2110-1**	*insO1, intA, tnpA*	*dnaE, dnaG, dnaQ_2, parB_3, polA, recA, repA, rnhA, uvrB_2*	*cor*, *smc*	*hsdR*	*dut*	cn_3039_IS26, IS26, IS911
**pEC2110-2**				*exc1*		
**pECV01-1**	*intA*	*dnaE, dnaG, dnaQ_2, holE_2, parB_3, polA, recA_1, repA, rnhA, umuD_5, uvrB_2*	*cor*, *smc*	*hsdM, hsdR*		
**pECV01-2**	*int, intM, ltrA, tnp, tnp1, tnpA, tnpR*	*impA-B, parA-B*		*vagC, vapC*		cn_2340_IS903, IS1R, IS1X3, IS26, IS903, ISEc15, TnpA_Tn3, TnpA_TnAs1, TnpR_Tn3, TnpR_TnAs1

## Data Availability

The original contributions presented in the study are included in the article/Appendix A, further inquiries can be directed to the corresponding author.

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
