# Peer review of "Whole-Genome Sequence Analysis of Antibiotic Resistance, Virulence, and Plasmid Dynamics in Multidrug-Resistant E. coli Isolates from Imported Shrimp"

_foods, 2024, doi:10.3390/foods13111766_

Round 1

Reviewer 1 Report

Comments and Suggestions for Authors

This manuscript # foods-3002867 contains a report on ‘Whole Genome Sequence Analysis of
Antibiotic Resistance, Virulence, and Plasmid Dynamics in Multidrug Resistant E. coli Isolates from
Imported Shrimp’. The authors used WGS analysis of antibiotic resistance and virulence traits in
MDR E. coli isolated from imported shrimp and found multiple resistance and virulence genes
distributed in the chromosomes and plasmids of the strain. It can be seen that the authors have
accomplished a relatively meaningful work, but there are still some detailed errors that need to be
corrected. It can be seen that the authors have accomplished a relatively worthwhile endeavour, but
there are still some detailed errors that need to be corrected.
(1) The distinction between chromosomal and plasmid sequences of the strains is not seen in the
BioProject number PRJNA802087, and the manuscript does not indicate the number of
plasmids carried by each strain and related information.
(2) Gene names and insertion sequence names should be in italics in the figures. Please check this
throughout the manuscript.
(3) How is plasmid mobility identified in Table 4? Is not reflected in the methodology.
(4) How was the replicon typing information for the plasmids in Table 4 determined? It is not
stated in the methodology.
(5) Line140-141, “Pangenome analysis of the plasmids revealed a high degree of similarity;
however, a few plasmids were observed with extra DNA length that includes some genes (Fig.
3).” How can the long segments of plasmid DNA be observed in this sentence? Figure 3 shows
a Pan-genome mapping of E. coli genomes, which also does not show any plasmid-related
markers.
(6) Line159-161, “Analysis of genomic islands (GI) and prophages across E. coli isolates showed
distinct counts of GIs and prophages, each linked to a unique combination of antibiotic
resistance and virulence genes (Table 2).” The argument that follows this sentence does not
seem to support this conclusion. In the Table 2, there is no clear pattern in the differences in
the number of GIs and prophages and in the carriage of antibiotic resistance genes and
virulence genes.
(7) The scale of genetic distances in Figure 2 is incomplete, please double check for changes.
(8) The discussion did not address any of the removable elements, and the relevant discussion
could be appropriately added.
(9) Line447-448, Latin names of strains need to be italicised. Please double check throughout the
manuscript.

Author Response

Response to Reviewers Comments

Dear Reviewers:

The authors extend their gratitude to the reviewer for their thorough examination of the manuscript and the insightful remarks that have enhanced its quality. Each valuable comment provided by the reviewer has been carefully considered and addressed through point-to-point responses in the revised manuscript. The concerns raised by the reviewers have been meticulously incorporated into the revised version, with line numbers based on the updated manuscript. We trust that our revisions have elevated the manuscript to a level that meets your satisfaction.

Reviewer #1

This manuscript # foods-3002867 contains a report on ‘Whole Genome Sequence Analysis of
Antibiotic Resistance, Virulence, and Plasmid Dynamics in Multidrug Resistant E. coli Isolates from Imported Shrimp’. The authors used WGS analysis of antibiotic resistance and virulence traits in MDR E. coli isolated from imported shrimp and found multiple resistance and virulence genes distributed in the chromosomes and plasmids of the strain. It can be seen that the authors have accomplished a relatively meaningful work, but there are still some detailed errors that need to be corrected. It can be seen that the authors have accomplished a relatively worthwhile endeavour, but there are still some detailed errors that need to be corrected.

(1) The distinction between chromosomal and plasmid sequences of the strains is not seen in the BioProject number PRJNA802087, and the manuscript does not indicate the number of plasmids carried by each strain and related information.

Response: Thank you for raising this important point. You're absolutely correct – the distinction between chromosomal and plasmid sequences wasn't clear from BioProject number PRJNA802087. Initially, only raw sequence reads and assembled sequences were submitted to NCBI. As a result, the BioProject number PRJNA802087 lacks the distinction between chromosomal and plasmid sequences for the strains. However, further in-depth analysis of the whole genome sequencing (WGS) data revealed the presence of plasmid sequences, which harbored genes associated with antibiotic resistance and virulence. Tables 4 and 5 indicates the number of plasmids carried by each strain and related information (antimicrobial resistance and virulence genes, replicon type, mobility, sizes, mobile elements).

(2) Gene names and insertion sequence names should be in italics in the figures. Please check this throughout the manuscript.

Response: Thank you for your meticulous attention to detail. You're absolutely right – gene and insertion sequence names throughout the manuscript should be italicized for clarity. We have carefully reviewed all figures and ensured that gene names and insertion sequence names are consistently formatted in italics.

(3) How is plasmid mobility identified in Table 4? Is not reflected in the methodology.

Response: Thank you for pointing out this important detail. You're absolutely correct – the methodology for identifying plasmid mobility was not previously described. VRprofile2 is a tool designed to detect mobile genetic elements within bacterial genome sequences. It furnishes information on mobility, Inc group, mobile genetic elements, antibiotic resistance genes, and virulence genes. A method for discerning plasmid mobility has been included in the Methods section (L99-100).

(4) How was the replicon typing information for the plasmids in Table 4 determined? It is not
stated in the methodology.

Response: Thank you for your keen eye. You're right, the methodology for determining replicon typing of the plasmids in Table 4 was missing. VRprofile2 has the capability to detect replicon typing. A method for discerning plasmid mobility has been incorporated into the Methods section (L99-100).

(5) Line140-141, “Pangenome analysis of the plasmids revealed a high degree of similarity;
however, a few plasmids were observed with extra DNA length that includes some genes (Fig.
3).” How can the long segments of plasmid DNA be observed in this sentence? Figure 3 shows
a Pan-genome mapping of E. coli genomes, which also does not show any plasmid-related
markers.

Response: We thank the reviewer for identifying this error. The text has been revised in L152-154 to more accurately reflect the findings: "Pangenome analysis revealed a high degree of similarity among most plasmids. However, a small number contained additional DNA sequences incorporating specific genes (Fig. 4A-B)."

(6) Line159-161, “Analysis of genomic islands (GI) and prophages across E. coli isolates showed
distinct counts of GIs and prophages, each linked to a unique combination of antibiotic
resistance and virulence genes (Table 2).” The argument that follows this sentence does not
seem to support this conclusion. In the Table 2, there is no clear pattern in the differences in
the number of GIs and prophages and in the carriage of antibiotic resistance genes and
virulence genes.
Response: Thank you for your careful review and for identifying this inconsistency. You're absolutely right – the initial wording in L159-161 did not accurately reflect the data presented in Table 2. We have revised the text (L171-173) to more accurately represent the findings: "E. coli isolates exhibited varying numbers of genomic islands (GIs) and prophages. A clear pattern linking the number of GIs/prophages to the carriage of antibiotic resistance and virulence genes was not evident in Table 2."

(7) The scale of genetic distances in Figure 2 is incomplete, please double check for changes.
Response: Thank you for your careful review of Figure 2. We appreciate you bringing this to our attention. We have re-examined the scale of genetic distances in Figure 2 and can confirm that it accurately reflects the calculated Average Nucleotide Identity (ANI) values between the strains. Please find the ANI statistics for your reference.

EC0002

EC110

EC1110

EC119

EC120

EC123

EC126

EC15

EC2110

EC331

EC335

EC338

EC339

ECV01

EC0002

100.00

100.00

100.00

99.97

99.97

99.99

99.99

98.41

100.00

99.99

100.00

100.00

100.00

98.52

EC110

100.00

100.00

100.00

100.00

100.00

99.99

99.99

98.44

100.00

99.99

100.00

100.00

99.99

98.54

EC1110

100.00

100.00

100.00

100.00

99.97

99.99

99.99

98.44

100.00

99.99

100.00

100.00

100.00

98.55

EC119

99.97

100.00

100.00

100.00

100.00

99.96

99.97

98.40

99.98

99.96

100.00

100.00

100.00

98.50

EC120

99.97

100.00

99.97

100.00

100.00

99.96

99.97

98.40

99.98

99.96

100.00

100.00

99.97

98.53

EC123

99.99

99.99

99.99

99.96

99.96

100.00

100.00

98.45

99.99

100.00

99.96

99.96

100.00

98.53

EC126

99.99

99.99

99.99

99.97

99.97

100.00

100.00

98.41

100.00

100.00

100.00

100.00

100.00

98.51

EC15

98.41

98.44

98.44

98.40

98.40

98.45

98.41

100.00

98.39

98.42

98.40

98.42

98.45

99.13

EC2110

100.00

100.00

100.00

99.98

99.98

99.99

100.00

98.39

100.00

100.00

100.00

99.98

100.00

98.49

EC331

99.99

99.99

99.99

99.96

99.96

100.00

100.00

98.42

100.00

100.00

100.00

99.96

100.00

98.53

EC335

100.00

100.00

100.00

100.00

100.00

99.96

100.00

98.40

100.00

100.00

100.00

100.00

100.00

98.51

EC338

100.00

100.00

100.00

100.00

100.00

99.96

100.00

98.42

99.98

99.96

100.00

100.00

100.00

98.50

EC339

100.00

99.99

100.00

100.00

99.97

100.00

100.00

98.45

100.00

100.00

100.00

100.00

100.00

98.51

ECV01

98.52

98.54

98.55

98.50

98.53

98.53

98.51

99.13

98.49

98.53

98.51

98.50

98.51

100.00

(8) The discussion did not address any of the removable elements, and the relevant discussion
could be appropriately added.
Response: Thank you for your insightful suggestion. You're right, the discussion currently lacks a specific focus on mobile elements. We have addressed this by incorporating a new section within the discussion (L348-371).

(9) Line447-448, Latin names of strains need to be italicised. Please double check throughout the manuscript.

Response: Thank you for your valuable comments and suggestions. We have carefully reviewed the manuscript and ensured that the Latin names of strains are italicized throughout as per your recommendation. Your attention to detail is greatly appreciated.

Reviewer 2 Report

Comments and Suggestions for Authors

Dear Authors, 

The manuscript needs some revisions.

First of all, the references in the text and in the "References" section are not formatted as required by the guidelines for authors, e.g. the DOI should not be given in this section; some references are given as "et al." in the list of authors, which is not correct in this section. Check italic for bacterial species in the Reference. I ask you to reformat correctly after reading the instructions for authors.

In section 2.1, the number of samples analysed was not reported, please specify how many samples were processed.

In paragraph 2.3, the number of strains sequenced was not given, please add this. 

In section 2.4, from lines 82 to 93, there are several omissions: the authors must add all the versions of the tools used, the date of accession and the corresponding hyperlink.

From a methodological point of view, the author should also carry out cgMLST and not only MLST, as it is more informative, has a higher resolution power and could also allow a phylogenetic correlation between the sequenced strains to be inferred.

In the 'Results' section, no information was provided on the amount of data generated and the quality of the reads and assembly used for the analysis. 

Line 97: Check format

Line 244-247: The classic MLST scheme is not discriminatory enough to state "potential clonality or a common evolutionary lineage", please perform cgMLTS to clarify the clonality. 

In my opinion, not all figures and tables should be included in the manuscript, some reported data should be included in "Supplementary materials".

Author Response

Response to Reviewers Comments

Dear Reviewers:

The authors extend their gratitude to the reviewer for their thorough examination of the manuscript and the insightful remarks that have enhanced its quality. Each valuable comment provided by the reviewer has been carefully considered and addressed through point-to-point responses in the revised manuscript. The concerns raised by the reviewers have been meticulously incorporated into the revised version, with line numbers based on the updated manuscript. We trust that our revisions have elevated the manuscript to a level that meets your satisfaction.

Reviewer #2

Dear Authors, 

The manuscript needs some revisions.

First of all, the references in the text and in the "References" section are not formatted as required by the guidelines for authors, e.g. the DOI should not be given in this section; some references are given as "et al." in the list of authors, which is not correct in this section. Check italic for bacterial species in the Reference. I ask you to reformat correctly after reading the instructions for authors.

Response: We sincerely appreciate your attention to detail. You're absolutely correct – the reference formatting needed adjustments to align with the journal's "Instructions for Authors." We have meticulously reviewed all references within the text and in the "References" section and have made the necessary corrections accordingly. Thank you for highlighting this issue.

In section 2.1, the number of samples analysed was not reported, please specify how many samples were processed.

Response: Thank you for pointing out this crucial oversight. You're absolutely correct – explicitly stating the number of samples analyzed is essential. In response, we have amended Section 2.1 (L58-59) to include the following sentence: "The 330 frozen, imported shrimp was thawed and incubated with Luria broth (Thermo Fisher Scientific, Waltham, MA) overnight." This addition ensures clarity and completeness in our reporting. We appreciate your diligence in reviewing our work.

In paragraph 2.3, the number of strains sequenced was not given, please add this. 

Response: Thank you for your thorough examination and for bringing this missing detail to our attention. We have now included the number of sequenced strains in Section 2.3 (L71-72), which reads: "Genomic DNA was extracted from overnight cultures of 14 MDR E. coli strains using the DNeasy Blood and Tissue Kit (Qiagen, Valencia, CA)." This addition ensures clarity and completeness in our methodology. We sincerely appreciate your diligence in reviewing our work.

In section 2.4, from lines 82 to 93, there are several omissions: the authors must add all the versions of the tools used, the date of accession and the corresponding hyperlink.

Response: Thank you for your meticulous review and for highlighting this important aspect. You're absolutely correct – providing comprehensive information about the software tools used in Section 2.4 is crucial. In response, we have updated Section 2.4 (L81-106) to include details about the versions, accession dates, and corresponding software names. This ensures transparency and reproducibility in our methodology. We appreciate your attention to detail and your valuable input.

From a methodological point of view, the author should also carry out cgMLST and not only MLST, as it is more informative, has a higher resolution power and could also allow a phylogenetic correlation between the sequenced strains to be inferred.

Response: Thank you for highlighting the importance of cgMLST. We value your suggestion and acknowledge its superior resolution for strain differentiation. The cgMLST data have been generated, presented in Table 1, and referenced in L267-270 to ensure clarity and completeness in our findings. Your input is greatly appreciated.

In the 'Results' section, no information was provided on the amount of data generated and the quality of the reads and assembly used for the analysis. 

Response: We sincerely appreciate your valuable comments and suggestions. The detailed information regarding WGS has been reported in the Microbiology Resource Announcements by Alotaibi et al. (2023) titled "Draft Genome Sequences of 14 Fluoroquinolone-Resistant Escherichia coli Isolates from Imported Shrimp," which we have referenced accordingly. Additionally, we have updated L77-79 as follows: "Detailed information of WGS was reported by Alotaibi et al. 12 and sequencing data are available under the National Center for Biotechnology Information’s (NCBI) BioProject number PRJNA802087." Thank you for your attention to detail and for guiding us in improving the clarity of our manuscript.

Line 97: Check format

Response: Thank you for highlighting the need for clarification on formatting. We have revised L106-109 as follows: “On the CGView server, Basic Local Alignment Search Tool (BLAST) analysis was performed using GenBank files of plasmid sequences, with an E value < 1e-10, an alignment length cutoff value of 100, and a percent identity cutoff value of 80.”

Line 244-247: The classic MLST scheme is not discriminatory enough to state "potential clonality or a common evolutionary lineage", please perform cgMLTS to clarify the clonality. 

Response: Thank you for your suggestion regarding cgMLST. We appreciate your emphasis on its higher resolution for strain differentiation. The cgMLST data have been generated, presented in Table 1, and referenced in L267-270 to ensure clarity and completeness in our findings. Your input is greatly valued.

In my opinion, not all figures and tables should be included in the manuscript, some reported data should be included in "Supplementary materials".

Response: Thank you for your suggestion regarding the placement of figures. We appreciate your feedback on manuscript conciseness. We have moved Figures 7A-E and 8A-B to the supplementary materials.
